# A genetically defined asymmetry underlies the inhibitory control of flexor–extensor locomotor movements

Olivier Britz[1], Jingming Zhang[1], Katja S Grossmann[1], Jason Dyck[2], Jun C Kim[3], Susan Dymecki[3], Simon Gosgnach[2], Martyn Goulding[1]*

[1]Molecular Neurobiology Laboratory, Salk Institute for Biological Studies, La Jolla, United States; [2]Department of Physiology, University of Alberta, Edmonton, Canada; [3]Department of Genetics, Harvard Medical School, Boston, United States

**Abstract** V1 and V2b interneurons (INs) are essential for the production of an alternating flexor–extensor motor output. Using a tripartite genetic system to selectively ablate either V1 or V2b INs in the caudal spinal cord and assess their specific functions in awake behaving animals, we find that V1 and V2b INs function in an opposing manner to control flexor–extensor-driven movements. Ablation of V1 INs results in limb hyperflexion, suggesting that V1 IN-derived inhibition is needed for proper extension movements of the limb. The loss of V2b INs results in hindlimb hyperextension and a delay in the transition from stance phase to swing phase, demonstrating V2b INs are required for the timely initiation and execution of limb flexion movements. Our findings also reveal a bias in the innervation of flexor- and extensor-related motor neurons by V1 and V2b INs that likely contributes to their differential actions on flexion–extension movements.

## Introduction

Terrestrial animals use their limbs to generate a broad array of motor behaviors, from stereotypical movements that include protective reflexes and locomotion to complex volitional tasks that are exemplified by reaching and grasping movements (*Grillner, 1975*; *Alstermark and Isa, 2012*). Charles *Sherrington (1906)* first demonstrated that all such motor behaviors rely on the reciprocal actions of flexor and extensor muscles around each limb joint and that terrestrial animals require reciprocal inhibition to move and articulate their limbs. It is known that reciprocal flexor–extensor motor activity is produced by inhibitory neurons in the spinal cord, many of which appear to be core components of the locomotor central pattern generator (CPG) (reviewed in *Kiehn, 2006*; *Goulding, 2009*; *Grillner and Jessell, 2009*; *Arber, 2012*). However, the functional organization of the inhibitory circuits that control flexor–extensor activity and the contribution that different inhibitory neuron cell types make to flexor–extensor motor control is still poorly understood.

Studies in the cat have identified a number of physiologically defined IN cell types that are candidates for exercising flexor–extensor control. The most prominent of these are reciprocal Ia inhibitory interneurons (IaINs), which are activated by muscle spindle afferents and inhibit antagonist motor neurons. IaINs are rhythmically active during locomotion (*Feldman and Orlovsky, 1975*; *Pratt and Jordan, 1987*) and scratching (*Deliagina and Orlovsky, 1980*). Peak IaIN activity coincides with the phase in which antagonist motor neurons are hyperpolarized (*Pratt and Jordon, 1987*; *Geertsen et al., 2011*). This activity profile is strong evidence of a central role in reciprocal inhibition. The contribution that non-reciprocal inhibitory Ib interneurons (IbINs) make to locomotion is less clear. While the IbINs that are innervated by Golgi tendon organs (GTOs) generally inhibit homonymous motor neurons under non-locomotor conditions (*Jankowska, 1992*; *Pearson and Collins, 1993*),

**\*For correspondence:**
goulding@salk.edu

**Competing interests:** The authors declare that no competing interests exist.

**eLife digest** Although there are many different movements an animal can make with its limbs—from reaching to walking—they all basically involve two sets of muscles that act as opposing levers around each joint. 'Flexor' muscles contract to bend the limb, and 'extensor' muscles contract to extend the limb. When an animal is walking these two sets of muscles contract repeatedly, one after the other. Inhibitory neurons in the spinal cord coordinate these walking movements by preventing the flexor or extensor muscles from contracting at the same time. In 2014, researchers discovered that two groups of inhibitory neurons, known as the V1 and V2b interneurons, are essential for this alternating pattern of flexing and extending of the limbs of newborn mice. However, these experiments were not able to assess the particular contribution that the V1 and V2b neurons each make to limb movements.

Now, Britz et al.—including several of the researchers involved in the 2014 study—have used a sophisticated genetic technique in mice to investigate the role that each group of neurons plays separately. This involved introducing a gene into either the V1 or V2b neurons that makes them susceptible to being killed with the diphtheria toxin. Injecting the mice with diphtheria toxin selectively removed these cells from the regions of the spinal cord that controls hindlimb movements.

Britz et al. found that removing either group of neurons prevented the mice from walking normally. Eliminating the V1 neurons caused extreme flexing of the hindlimbs, revealing that the V1 neurons are needed to extend the limb by inhibiting the motor neurons that contract the flexor muscles. In contrast, the loss of V2b neurons caused exaggerated hindlimb extension, indicating that the V2b neurons inhibit the motor neurons that innervate extensor muscles.

Both the V1 and V2b groups of neurons contain a wide range of different cell types. Future studies will therefore need to explore how these different cells are involved in coordinating the motions involved in walking.

during locomotion the Ib pathway exerts an excitatory effect on extensor motor activity (*Conway et al., 1987*; *Pearson and Collins, 1993*; *Gossard et al., 1994*; *Angel et al., 2005*). Nonetheless, inhibition mediated by IbINs has been observed during the extension phase of walking in humans (*Shoji et al., 2005*). This inhibition is associated with the unloading of the limb, and it suggests that IbINs may contribute to the phase transition from stance (extension) to swing (flexion). The role that Renshaw cells play in shaping flexor–extensor locomotor activity appears to be more limited (*Pratt and Jordan, 1987*). While these cells are rhythmically active during locomotor activity, reducing Renshaw cell transmission by pharmacological or genetic blockade of cholinergic transmission changes the periodicity of the locomotor rhythm with little discernible effect on flexor–extensor alternation (*Noga et al., 1987*; *Myers et al., 2005*).

Genetic analyses in mice have identified a number of molecularly defined interneuron populations that provide inhibition to motor neurons (reviewed in *Goulding, 2009*; *Arber, 2012*). Among these are three prominent populations of ipsilaterally projecting inhibitory interneurons: dorsal Lbx1-derived inhibitory INs (*Gross et al., 2002*; *Tripodi et al., 2011*) and ventral V1 and V2b INs (*Saueressig et al., 1999*; *Sapir et al., 2004*; *Zhang et al., 2014*). The V1 and V2b IN populations are both heterogeneous, being made up of multiple physiological cell types including Renshaw cells (V1 INs), IaINs (V1 and V2b INs) and putative IbINs (V2b INs) (*Sapir et al., 2004*; *Alvarez et al., 2005*; *Zhang et al., 2014*), as well as other as yet unidentified inhibitory cell types. Recently, we have found that the composite activities of the V1 and V2b INs are required to secure flexor–extensor alternation in the in vitro neonatal spinal cord and in newborn mice (*Zhang et al., 2014*). This finding is consistent with our demonstration that cells with the features of IaINs are derived from both V1 and V2b INs and that disynaptic reciprocal inhibition is only abolished when both of these populations are functionally inactivated (*Wang et al., 2008*; *Zhang et al., 2014*).

The limited repertoire of motor behaviors that can be assayed using the in vitro neonatal spinal cord preparation, together with our inability to discern any marked differences in the function of V1 and V2b INs with respect to generating an alternating flexor–extensor motor rhythm in vitro, prompted us to examine the contribution that V1 and V2b INs make to motor control in awake behaving mice. In particular, we were interested in determining whether these two inhibitory

interneuron classes control discrete aspects of limb movement with regard to flexor–extensor-driven motor behaviors. Our results demonstrate a striking functional bias in the actions of V1 and V2b INs on flexor–extensor motor activity, whereby the selective ablation of V1 vs V2b INs results in hindlimb hyperflexion and hyperextension, respectively. This functional bias was observed both in air-stepping juvenile mice and in adult animals performing motor tasks such as over-ground walking. In analyzing the inhibitory inputs to motor neurons from these two interneuron populations, we find a genetically defined bias in V1 and V2b connectivity that likely underpins the differential effects on flexor–extensor motor activity. Specifically, a higher proportion of the inhibitory contacts on flexor motor neurons come from V1 INs as compared to extensor motor neurons, whereas V2b INs preferentially contact motor neurons that innervate extensor muscles.

## Results

Efforts to define the contribution that V1 and V2b INs make to locomotion in awake behaving mice have been thwarted by an inability to selectively manipulate these cells in the spinal cord. Although En1 and Gata3 selectively mark spinal V1 and V2b INs, respectively, both genes are expressed elsewhere in the midbrain, medulla and cerebellum, and in non-neuronal tissues (*Davis and Joyner, 1988*; *van Doorninck et al., 1999*). Consequently, genetic protocols that employ *En1* and *Gata3* regulatory sequences alone to silence or ablate V1 and V2b INs cause perinatal death (*Gosgnach et al., 2006*; *Zhang et al., 2014*). To circumvent this issue, we devised a tripartite intersectional genetic system that utilizes the combined activities of Cre recombinase and FlpO recombinase and allows us to selectively express the diphtheria toxin receptor (DTR; *Buch et al., 2005*) in V1 or V2b INs that are located in the caudal spinal cord (*Figure 1A*), thereby making them selectively sensitive to diphtheria toxin (DTX).

Two strains of genetically modified mice were generated to facilitate the restricted intersectional ablation of V1 and V2b INs in the caudal spinal cord. The first mouse, which harbors a Cre- and Flp-dependent *Mapt*-doublestop-DTR (*Mapt*$^{ds-DTR}$) knock-in allele, uses regulatory sequences in the *Mapt/Tau* gene to restrict diphtheria toxin receptor (DTR) expression to neurons (*Figure 1A*). The second mouse containing 9.5 kb of the human *Cdx2* promoter (*Hinoi et al., 2007*) targets FlpO expression to the caudal torso and spinal cord (*Figure 1A*). Crosses using *Cdx2-FlpO* transgenic mice demonstrated the efficient recombination of multiple Flp recombinase-dependent reporters in the lumbosacral spinal cord (*Figure 1B–F*). Most importantly, the *Cdx2-FlpO* transgene displayed little or no recombination in anterior neural structures including the cortex, basal forebrain, midbrain, pons/medulla, and cerebellum. In P15 *En1*$^{Cre}$; *Cdx2-FlpO*; *R26*$^{ds-nlslacZ}$ mice, we observed a caudal pattern of nlslacZ expression in V1 INs that spanned mid-cervical to sacral levels of the spinal cord (*Figure 1C*; data not shown). When a *R26*$^{ds-HTB}$ reporter mouse (*Stam et al., 2012*) was used to assess the extent of *En1*$^{Cre}$ and *Cdx2-FlpO* recombination in V1 INs, GFP reporter expression was activated in virtually all V1-derived cells at lumbosacral levels (*Figure 1E,F*). By contrast, fewer than 20% of the V1 INs at upper- to mid-cervical levels displayed Flp-mediated recombination (*Figure 1D,F*). Most importantly, GFP was not detected in other supraspinal populations of En1-derived neurons, demonstrating the *Cdx2-FlpO* transgene selectively and efficiently targets V1 INs in the caudal spinal cord when used in combination with *En1*$^{Cre}$.

The fidelity and extent of Cre- and Flp-mediated recombination was then tested in relation to the *Mapt*$^{ds-DTR}$ allele. Gene expression from the *Mapt* locus was monitored using the nlslacZ reporter located in the downstream Flp recombinase-dependent stop cassette (*Figure 1A*). In the absence of *Cdx2-FlpO*, the nlslacZ reporter faithfully recapitulated En1 (*Figure 1G*) and Gata3 (*Figure 1I*) expression in the nervous system at E12.5. Most importantly, *Mapt*-regulated β-gal expression was completely excluded from non-neuronal tissues. In embryos that carry the *Cdx2-FlpO* transgene in addition to either *En1*$^{Cre}$ or G*ata3*$^{Cre}$, β-galactosidase staining was completely eliminated from the caudal spinal cord, demonstrating *Cdx2-FlpO* efficiently removes the FRT-flanked nlslacZ stop cassette in caudal V1 and V2b INs, respectively (*Figure 1H,J*).

## V1 INs are selectively ablated following exposure of *En1*$^{Cre}$; *Cdx2-FlpO*; *Mapt*$^{ds-DTR}$ mice to diphtheria toxin

To determine the efficiency and specificity of this technique, we monitored V1 IN ablation in P1 *En1*$^{Cre}$; *Cdx2-FlpO*; *Mapt*$^{ds-DTR}$ mice treated with a single dose of DTX and analyzed 6 days later. P1 control

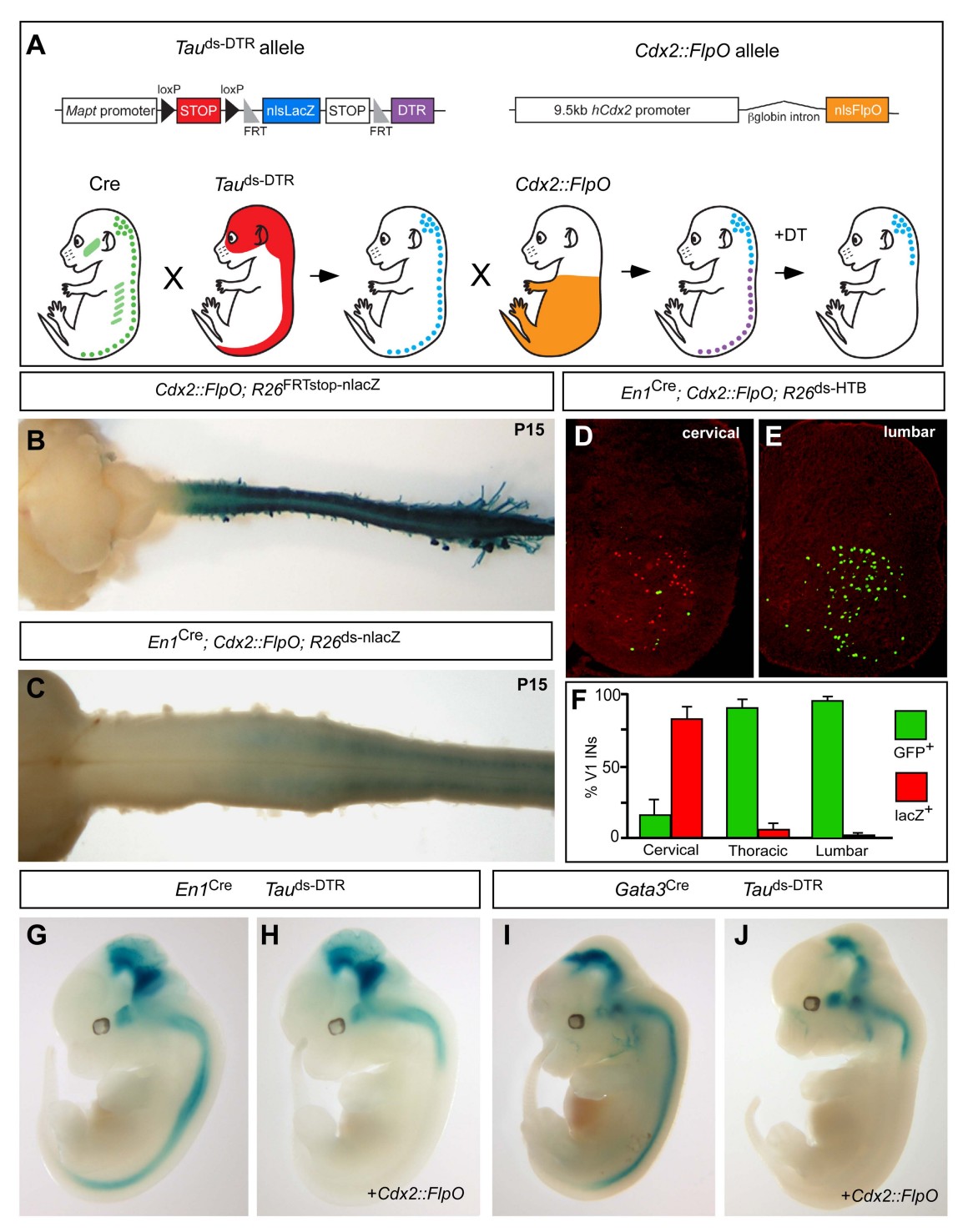

**Figure 1**. Genetic strategy for targeting DTR expression to V1 and V2b INs neurons in the caudal spinal cord. (**A**) Schematic of the tripartite genetic system used to restrict diphtheria toxin receptor (DTR) expression to caudal V1 and V2b INs. The *Tau*^ds-DTR and *Cdx2-FlpO* alleles are shown above. Cre-mediated recombination removes the first stop cassette to activate nuclear β-galactosidase (β-gal/nlsLacZ) expression in neurons (light blue). DTR (purple) is only expressed in neurons following Cre- and FlpO-mediated excision of both stop cassettes. (**B**) Whole mount image of a P15 *Cdx2-FlpO; R26*^FRTstop-nlslacZ brain and spinal cord viewed from the dorsal aspect showing β-galactosidase (β-gal) expression is restricted to the spinal cord and caudal PNS. (**C**) Whole mount image of the anterior spinal cord from a P15 *En1*^Cre; *Cdx2-FlpO; R26*^ds-lacZ mouse viewed from the ventral aspect showing expression of β-gal is restricted to caudal V1 INs and is absent at upper cervical levels and in the hindbrain. (**D–F**) Expression of a Cre- and FlpO-dependent GFP reporter (*En1*^Cre; *Cdx2-*

Figure 1. Continued

*FlpO*; *R26*^ds-HTB^) in V1 INs. Coronal sections from a *En1*^Cre^; *Cdx2-FlpO*; *R26*^ds-HTB^ spinal cord (**D**, **E**) in which V1 INs that have undergone Cre recombination alone express β-gal (red), and V1 INs cells that have undergone both Cre and FlpO recombinations express GFP (green). FlpO recombination is fully penetrant in lumbar V1 INs (**E**, **F**). (**G–J**) Images from E12.5 embryos showing that the *Mapt*^ds-DTR^ and *Cdx2-FlpO* alleles progressively restrict reporter gene expression to V1 and V2b INs in the caudal spinal cord. There is no β-gal expression in non-neuronal tissues. The absence of β-gal expression in the caudal CNS reflects the efficient excision of the nlsLacZ-stop cassette and expression of DTR in V1 and V2b INs.

mice (*En1*^Cre^; *Mapt*^ds-DTR^) were also treated with DTX and analyzed. Because En1 is no longer expressed postnatally, an *Ai6* Cre-dependent reporter allele (*Madisen et al., 2010*) was used to independently mark the V1 INs and verify their loss following DTX treatment. Whereas V1 INs were largely spared at upper cervical levels (*Figure 2A,B*), they were depleted by >90% in the thoracic, lumbar, and sacral spinal cord of *En1*^Cre^; *Cdx2-FlpO*; *Mapt*^ds-DTR^; *Ai6* mice (*Figure 2C,D*). Calbindin⁺ Renshaw cells, which are derived from En1⁺ progenitors (*Sapir et al., 2004*), were also largely missing from the caudal spinal cord (*Figure 2E,F*). By contrast, dorsal calbindin⁺ neurons that are not derived from En1⁺ progenitors were still present in normal numbers (*Figure 2—figure supplement 1*).

To confirm the specificity of V1 IN cell killing, sections from control and V1 IN-ablated cords were stained with an antibody to choline acetyltransferase (ChAT) to visualize motor neurons and cholinergic V0c INs. Both populations were unaffected by the DTR-dependent ablation of V1 INs (*Figure 2A–D*, data not shown). Moreover, there was no reduction in the number of Chx10⁺ V2a INs, a prominent population of excitatory neurons that are intermingled with V1 INs in lamina VII (*Figure 2G,H*). Further histological analysis revealed no change in the integrity of the spinal cord following V1 IN ablation (*Figure 2I,J*) nor was there any reduction in neuronal cell numbers (*Figure 2—figure supplement 1D*). We did observe a small transient increase in CD86 expression as is expected with the targeted killing of V1 INs (*Figure 2K,L*). This CD86 expression was localized to lamina VII where V1 INs reside, and in most instances co-localized with GFP-labeled V1 cell debris (*Figure 2L*, arrowhead). Most importantly, CD86 expression was not widespread nor was there any evidence of infiltration by CD45R-positive B cells or CD3-positive T cells (data not shown). These results agree with other studies showing DTR-mediated cell killing is highly selective and cell autonomous (*Buch et al., 2005*; *Hatori et al., 2008*).

A similar analysis was performed on *Gata3*^Cre^; *Cdx2-FlpO*; *Mapt*^ds-DTR^ mice post DTX-treatment (*Figure 2M–X*, *Figure 2—figure supplement 1E*). These mice also displayed a selective loss of V2b INs in the thoracic and lumbosacral spinal cord (*Figure 2R*). Other spinal neurons including motor neurons, V0c INs and Chx10⁺ V2a INs were completely spared in these cords (*Figure 2M–T*). Once again, we observed limited and localized expression of CD86 in lamina VII (*Figure 2X*). These data demonstrate that the intersectional ablation approach selectively and efficiently deletes V1 and V2b INs in caudal regions of the spinal cord.

## Ablation of V1 vs V2b INs has opposing effects on limb movements in juvenile mice

When DTX was administered to postnatal *En1*^Cre^; *Cdx2-FlpO*; *Mapt*^ds-DTR^ pups, it produced a strong hyperflexion phenotype within 3–4 days (*Figure 3*). Whereas control P7 pups flexed and extended their hindlimbs when suspended by their tails (*Figure 3A*), the hindlimbs of P7 DTX-treated *En1*^Cre^; *Cdx2-FlpO*; *Mapt*^ds-DTR^ V1 IN-ablated pups remained flexed (*Figure 3B*), even though their forelimbs were able to extend and display a full range of stepping movements (*Figure 3B*, asterisk). In contrast to control mice and V1 IN-ablated mice, P7 *Gata3*^Cre^; *Cdx2-FlpO*; *Mapt*^ds-DTR^ V2b IN-ablated mice maintained their hindlimbs in an extended state when suspended by their tail (*Figure 3C*), even though they were still able to flex and extend their forelimbs.

We then analyzed the hindlimb locomotor movements of DTX-treated mice that were induced to airstep by subcutaneous injection with L-DOPA. Kinematic analysis revealed strong rhythmic stepping in both the forelimbs and hindlimbs of control animals (*Figure 4*). By contrast, V1 IN-ablated animals displayed very little hindlimb movement, in contrast to the forelimbs that showed a normal range of motion. The hindlimbs of V1 IN-ablated animals, while occasionally displaying small rhythmic twitches (*Figure 4B*, asterisks), remained flexed during stepping (see *Figure 4A*, middle panel, arrowheads). Moreover, the maximal opening of the ankle joint in these animals was less than 80° compared to 145°

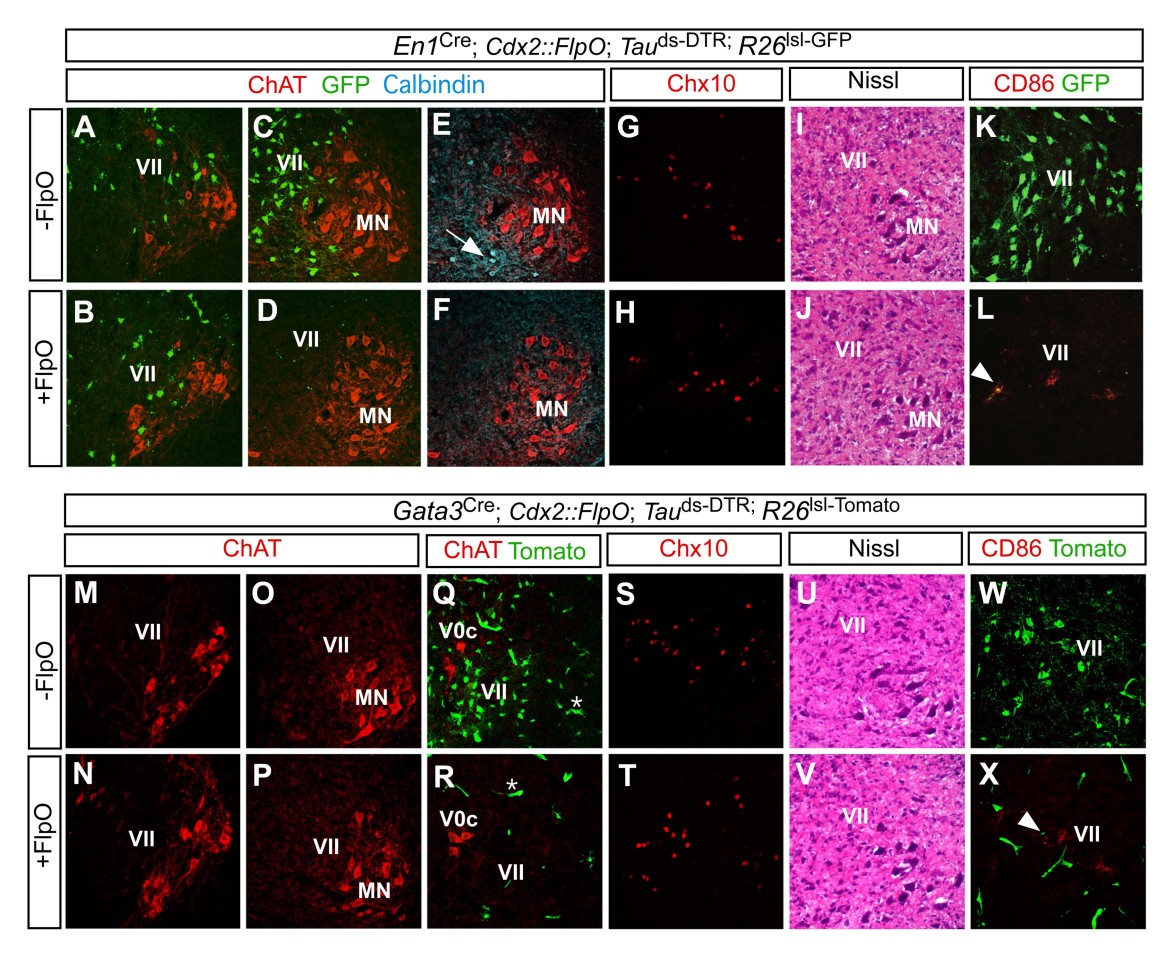

**Figure 2**. Restricted ablation of V1 and V2b INs following diphtheria toxin treatment. Immunohistochemical and histological analysis of P7 control, $En1^{Cre}$; $Cdx2\text{-}FlpO$; $Mapt^{ds\text{-}DTR}$ and $Gata3^{Cre}$; $Cdx2\text{-}FlpO$; $Mapt^{ds\text{-}DTR}$ animals 6 days after administering DTX. All sections are from the mid-lumbar cord except for those in panels **A**, **B**, **M**, and **N**, which are from the cervical cord. (**A–D**) En1-derived V1 INs (green) are selectively ablated in the lumbar spinal cord (c.f. **C**, **D**), whereas ChAT+ motor neurons (red) are spared at lumbar (**D**) and cervical levels (**B**). (**E**, **H**) Calbindin+ Renshaw cells (blue) are present in control cords (**E**, arrow) but not in V1 IN-ablated cords (**F**). (**G**, **H**) Chx10+ V2a INs are present in normal numbers following V1 IN-ablation. (**I**, **J**) Hematoxylin-eosin staining reveals no evidence of widespread neuronal cell loss or gliosis. (**K**, **L**) CD86+ microglia were occasionally observed in close proximity to V1 cell debris (arrowhead). (**M–P**) Motor neurons are spared at both cervical (**M**, **N**) and lumbar (**O**, **P**) levels following ablation of the V2b INs. (**Q**, **R**) V2b INs (green) are specifically deleted in V2b IN-ablated mice (**R**) while V0c neurons (red) are spared. (**S**, **T**) Chx10+ V2a INs are present in normal numbers following V2b IN-ablation. (**U**, **V**) Hematoxylin-eosin staining. (**W**, **X**) Localized CD86 expression (red) in lamina VII is associated with dying V2b INs (green).

The following figure supplement is available for figure 2:

**Figure supplement 1**. Selective ablation of ventral calbindin-expressing neurons.

for control animals (*Figure 4B,C*). In those instances where ankle joint did partially open, the overall change in angle was reduced from 90° to less than 30° (*Figure 4C*).

When V2b IN-ablated mice were induced to airwalk and analyzed in a similar manner, they displayed robust rhythmic forelimb stepping movements (*Figure 4*, right panels). By contrast, the hindlimbs of these mice remained extended and displayed very little in the way of flexion movements. Interestingly, L-DOPA also induced small rhythmic deflections of the hindlimb in these mice (*Figure 4B*, right panel, asterisk), indicating that the hindlimb locomotor CPG still produces an underlying locomotor rhythm. However, as the angle of the ankle joint was not reduced below 130° (*Figure 4C*), it appears that the hindlimb locomotor CPG is unable to elicit strong flexion movements in the absence of V2b inhibition.

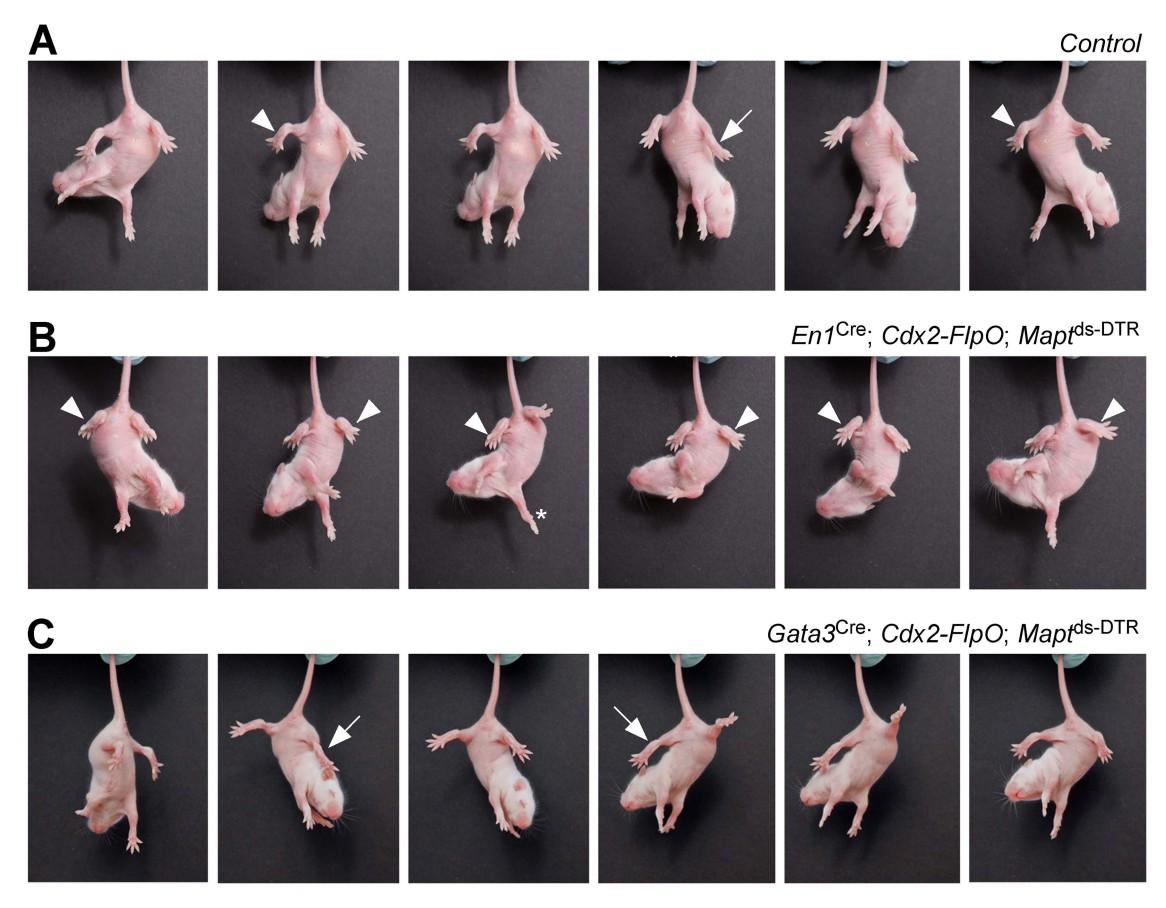

**Figure 3**. Mice lacking V1 and V2b INs show abnormal hindlimb movements. Time-lapse sequence images showing the hindlimb movements of P7 mice suspended by their tails. Mice were photographed 4 days after DTX treatment. The genotypes of the mice are indicated. Control mice were littermates that lacked the *Cdx2-FlpO* allele. (**A**) Control mice are able to flex (arrowheads) and extend (arrow) their hindlimbs when suspended by their tail. (**B**) Following the ablation of V1 INs, P7 mice lose their ability to extend their hindlimbs, which remain clasped to the body in a flexed position (arrowheads). The forelimbs of these mice are able to undergo extension movements (asterisk). (**C**) Ablation of V2b INs leads to pronounced extension of the hindlimbs (arrows) and impairment of hindlimb flexion movements.

To delve more deeply into the nature of the motor deficits that underlie the opposing motor phenotypes that arise when V1 and V2b INs are ablated, fine EMG recording electrodes were implanted unilaterally in the tibialis anterior (TA) and gastrocnemius (GS) muscles to measure ankle flexor and extensor motor activity during L-DOPA-induced air-stepping. Control mice displayed a regular alternating pattern of EMG activity in both muscles (*Figure 5A*) in which there was little or no overlap in TA and GS burst activity. This profile of TA-flexor and GS-extensor activity in control P7 mice is very similar to that seen in adult mice (see also *Figure 6*), where longer duration GS bursts are interspersed with short TA bursts. The GS extensor phase was also seen to increase proportionately with longer step periods, while the TA flexor burst period was constant across all stepping speeds (*Figure 5A,D*). This is in strong agreement with previous studies performed in rodents and cats (*Grillner, 1975*; *Halbertsma, 1983*; *Juvin et al., 2007*; *Frigon and Gossard, 2009*) showing that changes in duration of the extensor/stance phase are preferentially responsible for lengthening or shortening the step cycle.

V1 IN-ablated mice consistently displayed a marked increase in the duration of the TA burst, with the TA muscle remaining active for a proportionately longer period of each step cycle as compared to the GS muscle (*Figure 5B*). This finding is consistent with the idea that depleting V1 IN inhibition causes a preferential degradation of inhibition to TA flexor motor neurons. The increase in TA burst duration also caused a moderate lengthening of the step cycle period in V1 IN-ablated mice

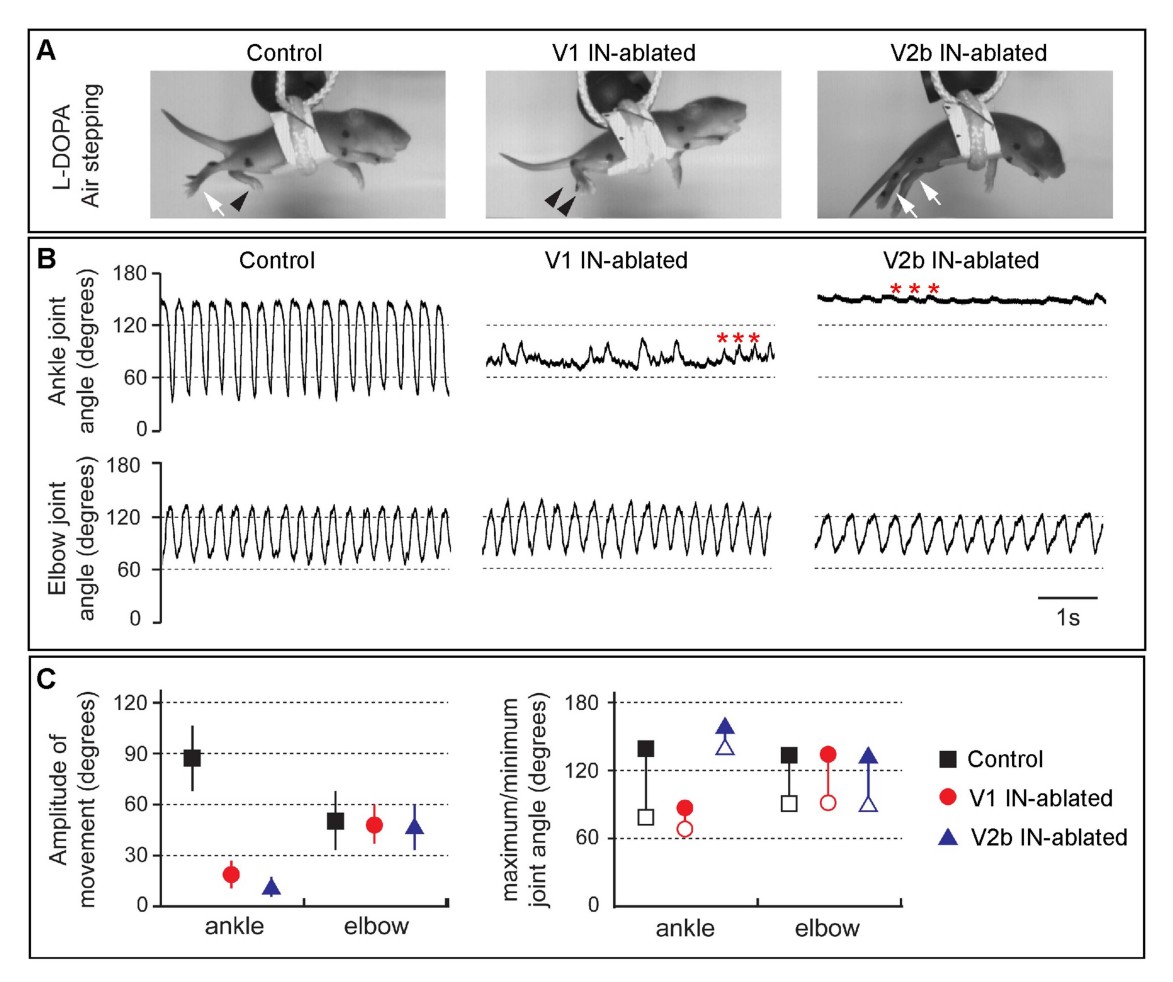

**Figure 4.** Juvenile mice lacking V1 and V2b INs display abnormal hindlimb movements. (**A**) Images of air stepping P7 animals following DTX injection. Flexed limbs are depicted with black arrowheads and extended limbs with white arrows. Control animals (left) flex and extend their hindlimbs. The hindlimbs of V1 IN-ablated animals (middle) remain flexed and those of V2b IN-ablated mice (right) are fully extended. (**B**) Traces showing the change in ankle (top) and elbow (bottom) joint angle during 5 s of airstepping. Deletion of V1 and V2b INs impairs movement of the ankle joint, whereas rhythmic movements are maintained for the elbow joint. Small rhythmic variations of the ankle joint angle occur in phase with movements of the elbow joint angle (asterisks). (**C**) Measured changes in joint movement showing the amplitude change (left), the maximal joint angle (filled symbol) and the minimal joint angle (open symbol) joint angle (right). The mean and s.d. is shown for 24 consecutive steps (n = 3 animals for each genotype).

(*Figure 5D*). V2b IN-depleted mice displayed a strikingly different pattern of EMG activity during airstepping. In addition to an aggregate slowing of the motor rhythm, the duration of GS extensor muscle activity was markedly elongated (*Figure 5C–E*) when compared to control and V1 IN-ablated animals. This slowing of the motor rhythm following the loss of V2b INs can largely be attributed to the increased duration of GS extensor activity, as the TA flexor bursts were similar in length to those seen in control mice (*Figure 5C,D*). Taken together, these findings demonstrate that a flexor-dominant motor rhythm emerges when V1 INs are deleted, while the loss of V2b INs gives rise to an extensor-dominant motor rhythm.

Further analysis of the EMG profiles of V1- and V2b-IN-ablated mice revealed a significant increase in the number and frequency of skipped or deleted GS and TA bursts in air-stepping juvenile mice (*Figure 5A*, arrowheads). Moreover, quantification of these deletions revealed a strong bias in their valency (*Figure 5D*). Whereas TA deletions were prevalent in V2b IN-ablated pups (*Figure 5A*, right panel, arrowheads), mice lacking V1 INs displayed a strong bias toward deletions in GS EMG activity (*Figure 5*, middle panel, arrowhead). The increased frequency of deletions in the V1- and V2b-IN-ablated mice (*Figure 5D*) coupled with the changes in phase duration (*Figure 5C*), strongly suggest that inhibition from

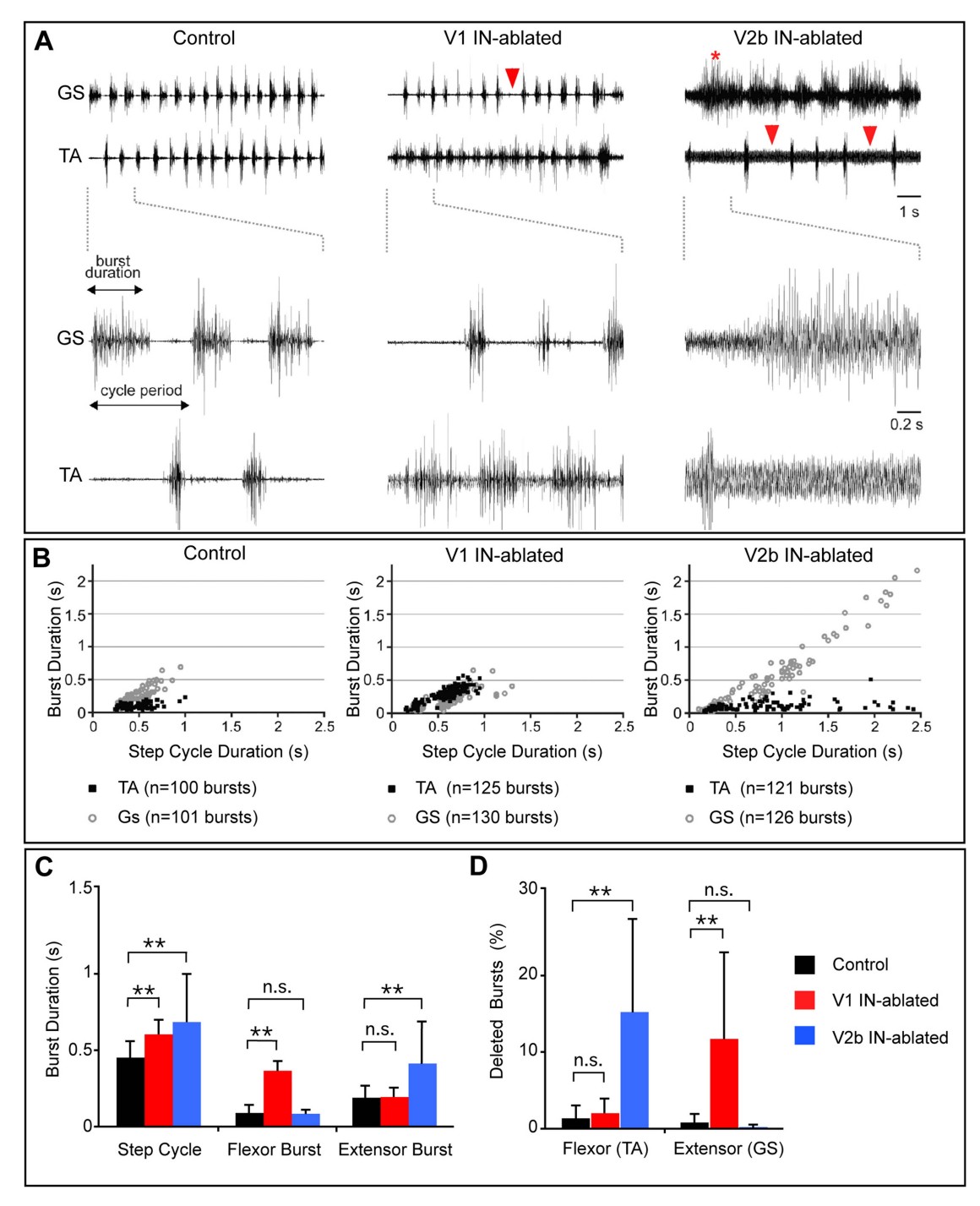

**Figure 5**. Altered rhythmic muscle activities during airstepping in the absence of V1 or V2b cells. (**A**) EMG recordings from the tibialis anterior (TA, flexor) and gastrocnemius (GS, extensor) muscles during L-DOPA-induced airstepping. Single deleted bursts are indicated by arrowheads. The asterisk marks a prolonged deletion that encompasses two step cycles. (**B**) Scatter plots show the relationship between ankle flexor (TA) burst duration and the step cycle period (black dots) and between ankle extensor (GS) burst duration and the step cycle period (gray circles). Each point represents the ratio of burst to step cycle duration for a single step. Control and V2b IN-ablated animals display an extensor dominant pattern, whereas V1 IN-ablated animals show a flexor-phase dominant pattern. Mean slope of regression calculations for each experimental group shows that the step cycle period in control mice (aGS = 0.72) is strongly correlated with GS burst duration, while TA bursts remain relatively constant (aTA = 0.12). The increase in step cycle duration in V1 IN-ablated mice is moderately correlated with flexor phase duration (aTA = 0.46). V2b IN-ablated animals display a pronounced increase in step cycle duration that is very highly correlated with the length of the extensor phase (aGS = 0.96). (**C**) Quantification of step cycle period, flexor and extensor burst duration for

*Figure 5. continued on next page*

Figure 5. Continued
control, V1 IN-ablated and V2b IN-ablated mice. (D) Percentage of skipped bursts/deletions as measured during twelve 10-s periods of locomotor activity
(n = 3 animals each).

V1 and V2b INs facilitates the transition from swing to stance and from stance to swing, respectively. Our observation that the deletions in both V1- and V2b-IN-ablated mice fall into the non-resetting category strongly suggests that both of these genetically defined cell populations are involved in pattern formation rather than rhythm generation (*Lafreniere-Roula and McCrea, 2005*).

## Deficits in flexor–extensor motor coordination persist following V1 and V2b IN ablation

In order to examine the nature of any persistent changes to locomotor behavior that arise from the loss of V1 and V2b INs, we used combined kinematics and EMG recordings to quantitatively analyze changes in the gait of adult animals 3–5 weeks post DTX treatment. By this time, V1- and V2b-IN-ablated mice had regained partial control of their hindlimbs (see 'Discussion') and were able to use them to bear weight. V1 IN-ablated mice still displayed marked deficits in their gait, including a persistent hyperflexion of the hindlimbs during walking, which was particularly prominent in the mid-swing and early stance phases of the step cycle (*Figure 6A,B*, middle panel). Furthermore, V1 IN-ablated mice flexed their hindlimbs faster during the swing phase indicating that they are no longer able to properly modulate the speed and/or force of their flexion movements (*Figure 6C*).

The EMG analysis of V1 IN-ablated mice revealed marked changes in the duration and co-activity of TA and GS muscle activity. These included (1) a persistent broadening of TA muscle activity, (2) an overlap in GS and TA burst activity during the transition from stance to swing, and (3) an increase in GS activity at the end of the stance phase (*Figure 6D*, middle panel, asterisk; *Figure 6—figure supplements 1, 2*). The increase in GS EMG activity during stance phase appears to be a late behavioral modification that facilitates the opening of the ankle joint during walking, as it is not seen during the early phase of V1 IN ablation (see *Figure 6—figure supplement 2*). The co-activation of the TA and GS muscles was even more pronounced during swimming with EMG analysis revealing synchronous TA and GS muscle activity (*Figure 6E*, middle panel, asterisk) that causes the extension of the hindlimbs to be aborted during the power stroke. One factor that may contribute to the synchronous flexor–extensor activity seen during swimming may be the reduction in sensory feedback from GTOs that has been shown to occur (*Akay et al., 2014*). In V1 IN-ablated mice that lack sensory feedback gated by the V1 INs, the likely loss of Ib-mediated sensory inhibition during swimming could account for the observed co-activation of flexor–extensor muscles and abortive hindlimb movements (*Figure 6*, *Figure 6—figure supplement 2*; see 'Discussion').

In addition to the hyperflexion phenotype, V1 IN-ablated mice displayed a prominent rear paw overshoot during walking (*Figure 6—figure supplement 3*). Kinematic analysis of walking V1 IN-ablated mice revealed two factors that contribute to rear paw overshoot: (1) an increase in the speed and degree of ankle flexion movements during swing (*Figure 6C*) and (2) an increase in the angular rotation of the pelvis during stepping (*Figure 6—figure supplement 3*). This rotation of the pelvis appears to be a behavioral adaption that helps extend the forward throw of the leg during late swing phase, so as to negate the reduced hindlimb extension that occurs when V1 INs are depleted from the spinal cord.

In contrast to V1 IN-ablated mice, the locomotor phenotype that persists in mice lacking V2b INs was much milder, with the most notable change being a prolongation of the stance phase that results in overextension of the ankle joint during walking (*Figure 6A*, right panel, arrowhead). EMG analysis revealed the likely cause of this delay, namely a second ectopic burst of GS motor activity as the limb is transitioning from stance to swing (*Figure 6D*, right panel, asterisk). The delay in initiating flexion was not observed when the mice were swimming nor did we detect any marked differences in TA and GS EMG activity between control and V2b IN-ablated mice during swimming (*Figure 6E*, right panel).

## The loss of V1 INs in adult mice results in the prolongation of flexor motor neuron activity

A further set of experiments were performed to probe the nature of the early motor deficits that arise when V1 INs are ablated in adult mice. Unfortunately, we were unable to perform a similar analysis of

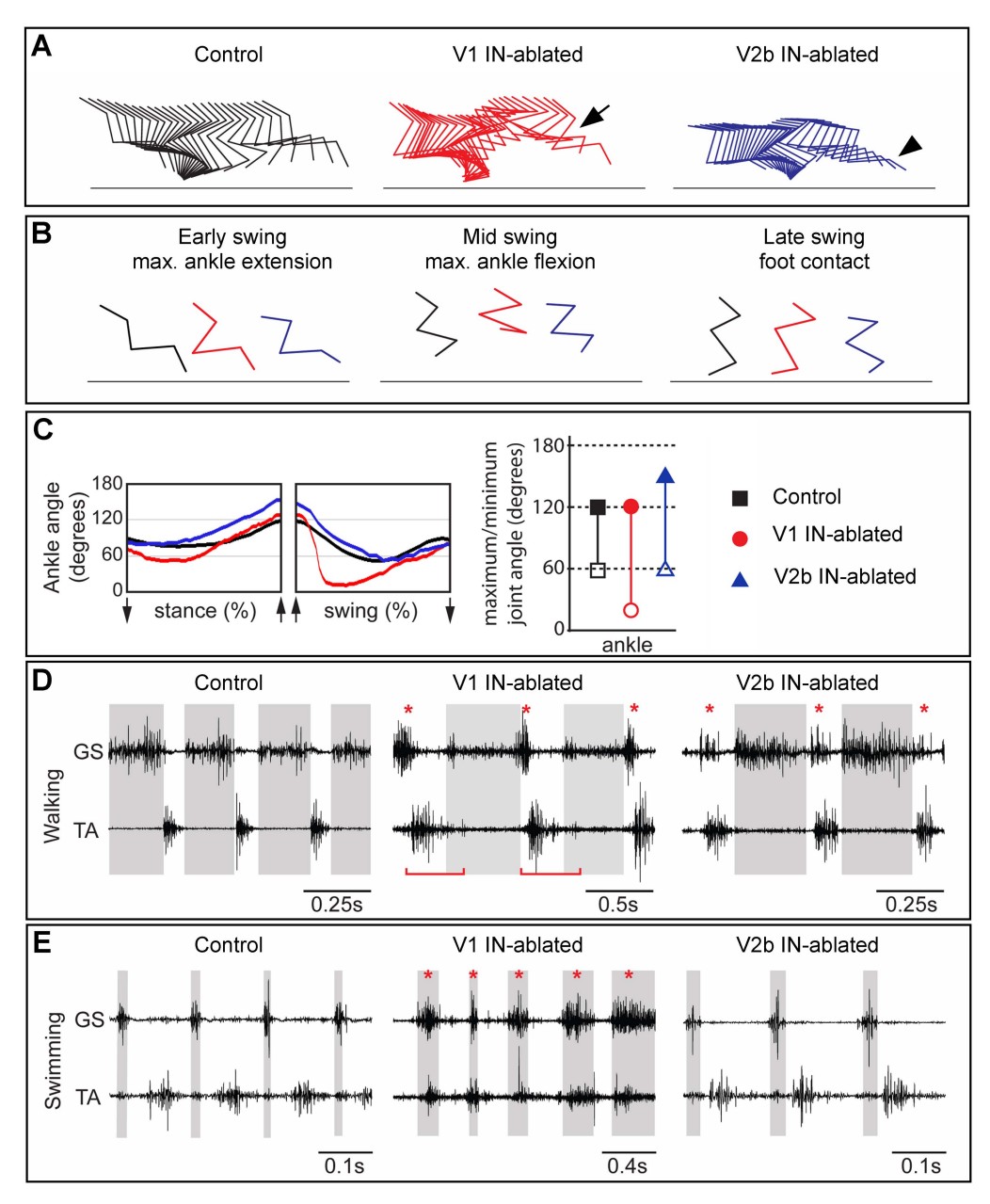

**Figure 6**. Mice lacking V1 and V2b INs have opposing changes to their gait. Hindlimb kinematics from control (left), V1 IN-ablated (middle), and V2b IN-ablated (right) mice at 3 weeks post DTX injection, by which time functional recovery was maximal. (**A**) Representative stick figure diagrams showing one complete step cycle (swing and stance) for the hindlimb. The arrow (middle) indicates hyperflexion during early swing phase in V1 IN-ablated mice. The arrowhead shows hyperextension of the ankle joint during late stance in V2b IN-ablated mice. (**B**) Comparison of limb positions at the transition from stance to swing (left), at mid-swing (middle), and at the swing to stance transition (right). (**C**) Representative angular changes to the ankle joint. The arrows indicate when the foot is lifted (stance to swing) and when the foot is planted (swing to stance). (**D**, **E**) Simultaneous EMG recordings of the GS and TA muscles in one leg during walking (**D**) and swimming (**E**). The bar in **D** indicates the expansion in TA activity. The asterisks indicate co-activation of the TA and GS muscles. Note the synchronous activity of both muscles during swimming in V1 IN-ablated mice.

The following figure supplements are available for figure 6:

**Figure supplement 1**. Phase relationship between TA and GS EMG activity.

*Figure 6. continued on next page*

*Figure 6. Continued*

**Figure supplement 2**. Progressive changes in EMG activity following V1 IN ablation.

**Figure supplement 3**. Altered limb and body movements in mice lacking V1 INs.

the early motor deficits that occur in adult V2b IN-ablated animals, as treating these animals with a single high dose of DTX caused bowel blockage and increased morbidity. This is likely to be due to the expression of Gata3 in a subset of enteric neurons. Whereas control mice (n = 5 animals) displayed a highly reproducible pattern of EMG activity both before and after DTX treatment, with the iliopsoas (IP) and TA muscles only being active during the swing phase of the step cycle (*Figure 6—figure supplement 2*, upper left panel), the depletion of V1 INs (n = 6 animals) resulted in ectopic IP and TA muscle EMG activity within 5 days of DTX treatment (*Figure 6—figure supplement 2B*, upper panel). This ectopic TA and IP muscle activity was even more pronounced 7 days after DTX treatment (*Figure 6—figure supplement 2C*, upper panel), with the TA muscle being co-active during stance with the GS muscle (see bars). A similar expansion of IP activity was observed, although to a less extent than that seen in the TA (*Figure 6—figure supplement 2C*, arrowhead). We posit that the co-activation of the TA and IP muscles during stance constitutes the underlying mechanism that restricts the hip and ankle joints from opening during walking, and it also contributes to the postural changes that occur in V1 IN-ablated mice following DTX treatment. By contrast, there was no increase in GS activity during the swing/flexor phase indicating the loss of V1 INs has an effect on ankle flexor but not extensor activity during walking.

A careful comparison of kinematic movements (*Figure 6—figure supplement 2*, lower panels) and the time locked EMG traces (*Figure 6—figure supplement 2*, upper panels) revealed a close correspondence between angular movement of the ankle and EMG activity in the TA and GS muscles. During the early phase of V1 IN cell loss (5 days p.i.), the ankle joint angle at maximal flexion was reduced from 40° to less than 30° (lower middle panel). There was also a progressive decrease in maximal extension of the ankle joint in early swing phase (*Figure 6—figure supplement 2*, lower panels, asterisks). A similar hyperflexion phenotype was noted for the knee and hip joints, as indicated by their reduced angular changes during walking (*Figure 6—figure supplement 2*, lower panels). In the case of the hip joint, this is consistent with the observed expansion of IP muscle activity (*Figure 6—figure supplement 2C*, upper panel, arrowhead). In summary, our findings reveal that V1 INs function to (1) restrict flexor motor activity and promote active limb extension during stance and (2) moderate flexion movements during the swing phase of the step cycle.

## Optogenetic activation of V2b INs preferentially suppresses extensor-related locomotor activity

The weakening of ankle extensor inhibition in V2b IN-ablated mice, as indicated by ectopic GS burst activity during the swing phase of the step cycle and hyperextension of the ankle during walking (*Figure 6*), led us to ask whether V2b INs preferentially inhibit extensor–motor activity. To test this, we took advantage of the neonatal spinal cord preparation, which can be induced to produce fictive locomotion in vitro (*Figure 7A*, *Kiehn, 2006*; *Goulding, 2009*) and is amenable to optogenetic manipulation (*Hagglund et al., 2010*). Spinal cords from P0-P1 *Gata3*^Cre; *R26*^lsl-ChR2 (*Ai32*) pups displayed a characteristic alternating pattern of L2 flexor-related and L5 extensor-related locomotor rhythm in the presence of NMDA and 5-HT (*Figure 7B–D*, n = 6/6 cords). However, upon activating channelrhodopsin in V2b INs, there was a strong and highly selective decrease in extensor-related motor activity as measured by extracellular recordings from the L5 ventral root (*Figure 7B–D*, bar). By contrast, L2 flexor-related motor activity remained largely unchanged following V2b IN activation, although there was some disturbance in the coherence of the motor rhythm. Activation of V1 INs had a more pronounced effect on the motor rhythm, in that it completely suppressed L2 flexor-related and L5 extensor-related rhythmic bursting (*Figure 7E*, bar, n = 5/5 cords). This is likely to be due to the activation of Renshaw cells, which are known to strongly inhibit motor neurons (*Windhorst, 1996*; *Bhumbra et al., 2014*). In summary, the suppression of extensor-related L5 motor activity in the in

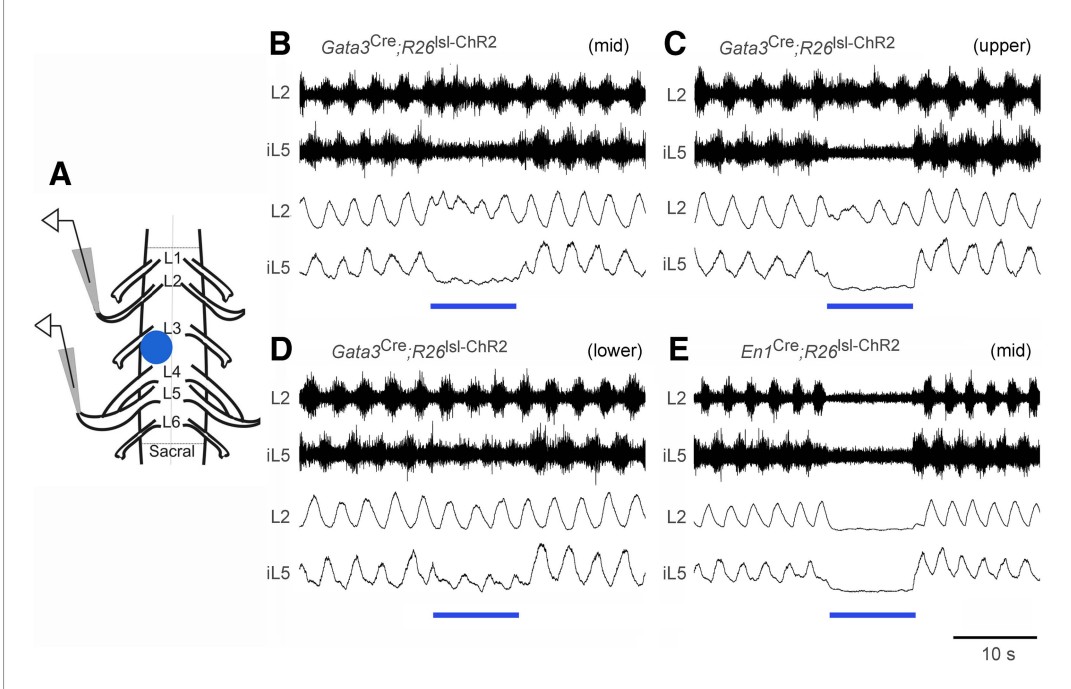

**Figure 7**. Effect of optogenetic activation of V2b and V1 INs on in vitro locomotion. (**A**) Schematic showing the in vitro recording setup used for the localized light activation of ChR2 in V1 and V2b INs. (**B–D**) Representative ENG recordings from a P0 *Gata3*^Cre^; *R26*^Isl-ChR2^ (*Ai32*) spinal cord showing the suppression of L5 extensor-related activity in the presence of blue light photostimulation (blue bar). The lumbar levels that were photostimulated are indicated as upper (L1–L2), mid (L3–L4), and lower (L5–L6). (**E**) Representative recording from a P0 *En1*^Cre^; *R26*^Isl-ChR2^ (*Ai32*) spinal cord showing suppression of L2 and L5 ventral root activity following photostimulation of V1 INs at L3–L4. Upper traces represent the raw filtered ENG recordings. Lower traces display the matching online rectified ENG signal.

vitro spinal cord preparation when V2b INs are activated provides strong evidence that V2b INs can functionally inhibit extensor-related motor activity during locomotion. This finding is consistent with the expansion of extensor motor neuron activity that we see following V2b IN ablation (*Figures 5, 6*).

## Hindlimb flexor and extensor motor pools are differentially innervated by V1 and V2b INs

In view of the opposing phenotypes that arise from ablating V1 and V2b INs, we asked if differences in the innervation of hindlimb extensor and flexor motor pools by these two classes of inhibitory neuron might contribute to their opposing actions on limb flexion and extension movements. To address this question, defined hindlimb motor pools were visualized by backfilling them from their respective muscles with Cholera Toxin-B (CTB). A conditional *Thy1*^Isl-YFP^ transgene reporter (*Buffelli et al., 2003*) was then used in combination with *En1*^Cre^ or *Gata3*^Cre^ to label the terminal processes of V1 and V2b INs. Presumptive inhibitory synapses on CTB-labeled motor neurons were then identified with antibodies to vGAT and GlyT2 (*Figure 8A*). The analysis of V1 and V2b inhibitory contacts focused on the soma and proximal dendrites of motor neurons, which is where inputs from inhibitory interneurons such as IaINs and Renshaw cells tend to be concentrated (*Jankowska and Roberts, 1972*; *Brown, 1981*).

Counts of V1-derived inhibitory contacts on motor neurons revealed a strong positive bias in their innervation of the TA ankle flexor motor neurons (*Figure 8*). This preferential innervation is also seen for hip flexor motor neurons, which receive a greater percentage of inputs from V1 INs when compared to their antagonist extensor motor pools (*Zhang et al., 2014*). In comparing V1 inhibitory inputs onto TA and GS motor neurons, we counted an approximate twofold greater number of V1-derived synaptic contacts onto TA motor neurons as compared to their antagonist GS counterparts (*Figure 8B,C*). Conversely, the proportion of inhibitory contacts onto GS (ankle)

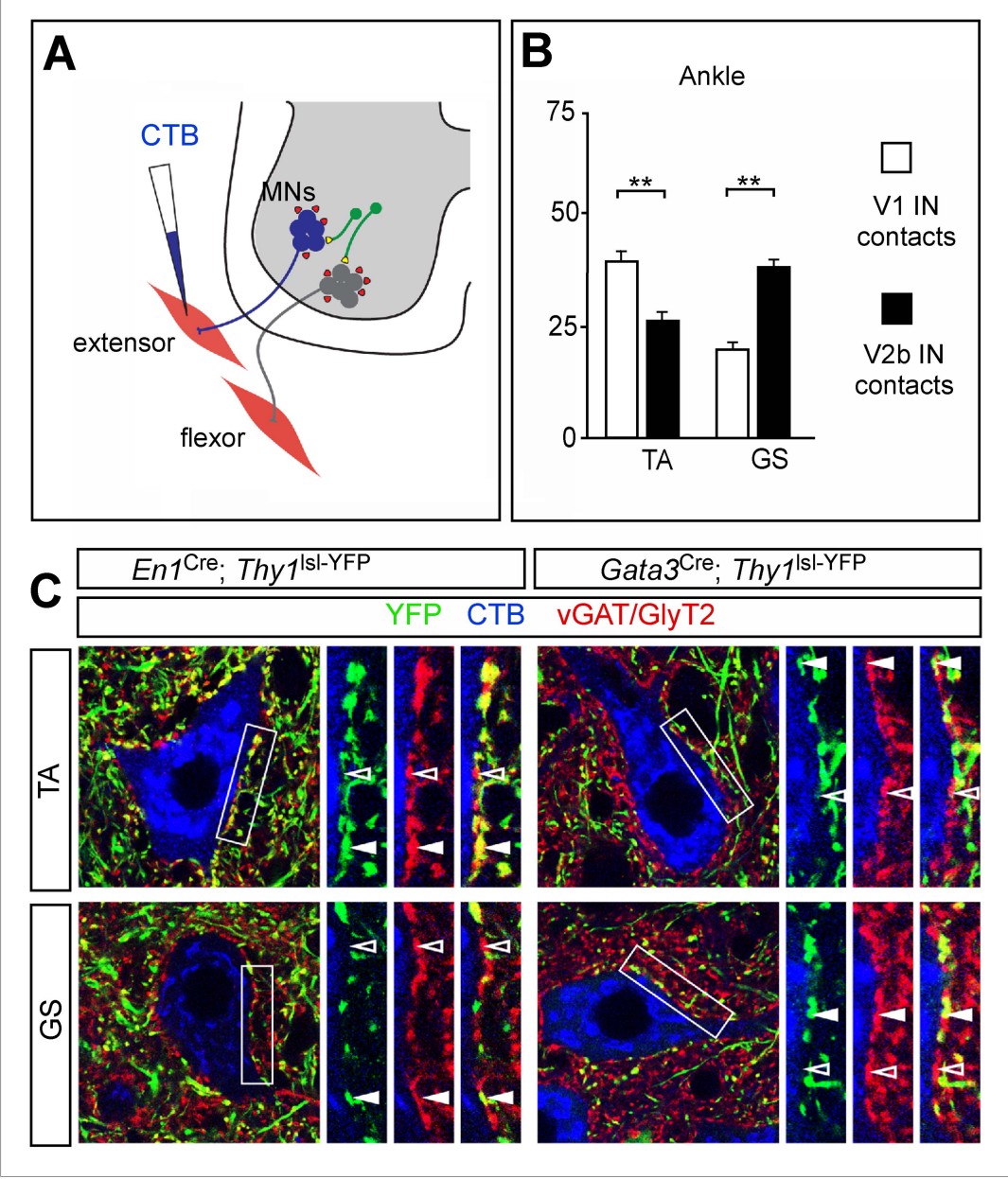

**Figure 8**. Biased V1 and V2b connections onto flexor- and extensor-related motor neurons. (**A**) Schematic of experimental design used for labeling V1 and V2b inhibitory contacts on specific motor pools. Motor neurons (MNs) were retrogradely labeled by injecting Cy5-CTB into individual hindlimb muscles at P11. Interneuron class-specific contacts were labeled with a conditional *Thy1*[Isl-YFP] reporter allele (green) and inhibitory contacts were detected with antibodies to vGAT and GlyT2 (red). (**B**) Putative V1- and V2b-derived inhibitory contacts (vGAT/GlyT2 (red) and Thy1-YFP reporter (green)) onto CTB-labeled TA and GS motor neurons (blue). Filled arrowheads indicate examples of inhibitory synaptic terminals that co-localize with YFP. YFP-negative inhibitory contacts are marked with open arrowheads. (**C**) Quantification of V1- or V2b-IN-derived inhibitory contacts on the soma/proximal dendrites of defined hindlimb motor neurons (n = 12 motor neurons per pool). A greater number of putative V1 IN inhibitory synapses contact flexor motor neuron pools as compared to their antagonist extensor-related motor pools. In contrast, V2b-derived contacts represent a greater proportion of the inhibitory synapses onto extensor-related motor neurons as compared to antagonist flexor-related motor neurons. Abbreviations: GS, gastrocnemius; TA, tibialis anterior. Error bars: mean ± s.d.

The following figure supplement is available for figure 8:

*Figure 8. Continued*

**Figure supplement 1**. Axonal projections of V1 and V2b INs.

extensor motor neurons from V2b INs was higher than the proportion of V2b contacts onto their TA counterparts (*Figure 8C*). We have not seen any significant difference in the proportion of V2b IN contacts onto the knee quadriceps (Q) and BF/St motor pools (*Zhang et al., 2014*). These two knee muscle groups are bifunctional (*Yakovenko et al., 2002*), and the near neutral weighting of V2b IN inputs onto these motor pools may reflect their bi-functional nature with regard to knee and hip flexion-extension movements. In summary, our findings reveal an overall asymmetry in V1 and V2b in inputs, with flexor-related motor pools receiving proportionately fewer inputs from V2b INs and more from V1 INs, while V2b INs exhibit a bias toward extensor-related motor pools.

## Discussion

This study defines, for the first time, the functional contribution that V1 and V2b INs make to flexor–extensor motor behaviors in awake behaving mice by showing that these two inhibitory neuron classes contribute to the functional antagonism of limb movements via their differential actions on limb flexor–extensor motor activity. V1 INs restrict flexor-related motor activity during the stance phase of the step cycle to promote full extension of the limb, whereas V2b INs limit extensor activity to facilitate normal leg flexion movements and the transition to swing. Our results also reveal that the control of limb flexion and extension movements by the V1 and V2b INs is likely to be exerted by graded rather than absolute differences in the inhibition of antagonist motor neurons by V1 and V2b INs.

### Generation of a flexible genetic system for analyzing neuronal function in the spinal cord

Multiple methods have been implemented to inactivate genetically defined neurons within a circuit (*Tan et al., 2006*; *Luo et al., 2008*; *Kim et al., 2009*). These include blocking neurotransmission with tetanus toxin (*Yu et al., 2004*; *Zhang et al., 2008*, *2014*) and suppressing neuronal excitability with heterologous chloride channels or G-protein-coupled receptors (GPCRs) that activate GIRK channels (*Gosgnach et al., 2006*; *Tan et al., 2006*; *Armbruster et al., 2007*; *Ray et al., 2011*). These systems all have drawbacks. Approaches using tetanus toxin are often non-inducible, while ligand-mediated silencing using Gi-coupled receptors can be highly variable and difficult to quantify. By contrast, DTR-mediated neuronal ablation has the advantage of being highly quantifiable, and the timing of cell killing can also be controlled to minimize potential developmental changes due to the chronic silencing or ablation of cells at early developmental times (*Gosgnach et al., 2006*; *Crone et al., 2008*; *Zhang et al., 2008*). Most importantly, by using an intersectional approach to restrict DTR expression to the caudal CNS, we were able to spare essential functions such as respiration and chewing. With further refinements, this intersectional system can be used to target and ablate discrete populations of neurons, including genetically defined subsets of V1 and V2b INs.

### Differential control of flexor and extensor movements by V1 and V2b INs

Our results demonstrate that V1 INs facilitate the transition from swing to stance, and V2b INs facilitate the transition from stance to swing. In the *Xenopus* tadpole, V1 (aIN) cells promote the transition between the active and inactive phases of the swimming rhythm by providing early phase inhibition to motor neurons (*Li et al., 2004*). The CiA cells, which are homologous to the V1 INs, are likely to function in a similar fashion in zebrafish (*Higashijima et al., 2004*). Interestingly, we see an increase in the incidence of GS (extensor) EMG deletions in airstepping mice that lack V1 INs, suggesting the transition from flexion to extension is compromised (*Figure 5*). Interestingly, these deletions, and the flexor deletions that arise from the loss of the V2b INs, resemble non-resetting deletions in the cat and turtle (*Grillner and Zangger, 1979*; *Lafreniere-Roula and McCrea, 2005*; *McCrea and Rybak, 2008*; *Stein, 2008*). The absence of resetting deletions suggests both populations belong to the pattern forming, rather than rhythm generating, layer of the locomotor

CPG. Moreover, the opposing nature of the deletions that occur when V1 vs V2b INs are inactivated suggests that the differential silencing of these two inhibitory cell populations may contribute to the valence of corrective reflexes such as the stumbling corrective reaction (*Forssberg, 1979*) and crossed extension reflex (*Sherrington, 1910*).

## Biased vs segregated inhibitory inputs to antagonist motor neurons

Our preliminary anatomical analysis of the organization of V1 and V2b inputs to motor neurons (*Figure 8*) reveals a genetically encoded bias in the innervation of flexor vs extensor motor pools by V1 and V2b INs (see also *Zhang et al., 2014*). Although we were unable to score inhibitory contacts on the distal dendrites of motor neurons, studies showing IaIN and Renshaw cell inhibitory synapses are preferentially located on the soma, and proximal dendrites of motor neurons (*Brown, 1981*; *Fyffe, 1991*) suggests that the counts we obtained in this study are likely to be a reasonable measure of the density of V1 and V2b inhibitory synaptic contacts on motor neurons. This in turn suggests that relative differences in the inhibitory drive to motor neurons from the V1 and V2b IN populations may contribute to the opposing actions of V1 and V2b INs on flexor–extensor movements. Our finding that ChR2 activation of V2b INs in the isolated spinal cord selectively suppresses L5 extensor-related motor neuron activity (*Figure 7*) is consistent with such a model.

In considering the differential innervation of motor neurons by V1 and V2b INs, we would like to suggest that there are multiple advantages in having a system where biased rather than segregated inhibitory inputs from V1 and V2b INs determine the valence of flexor–extensor movements. First, biased inputs from the V1 and V2b INs would facilitate the graded activation of synergist and antagonist motor neurons. This graded recruitment of antagonist flexor–extensor motor neurons would be expected to play an important role in modulating limb stiffness and compliance, which is necessary for smooth movements and postural control. Second, the co-innervation of flexor and extensor motor neurons by V1 and V2b INs would enable motor neurons to actively summate and compare inhibitory inputs that are differentially gated by these two inhibitory interneuron populations.

## Inhibition and the push–pull control of motor activity

There is growing evidence that motor neurons receive a mixture of tonic and dynamic inhibition during locomotion that together with excitatory inputs regulate motor neuron excitability (*Berg et al., 2007*; *Johnson et al., 2012*). In particular, the altered membrane conductances produced by concurrent excitation and inhibition are believed to be a fundamental mechanism for changing the gain and dynamic properties of neurons (*Chance et al., 2002*; *Mitchell and Silver, 2003*; *Abbott and Chance, 2005*). Consequently, gain modulation and increases in the variability of motor neuron spiking represent an important mechanism for altering the dynamic range of motor neuron activity, so as to produce smooth gradients of muscle force transduction (*Kristan, 2007*; *Johnson et al., 2012*).

*Johnson et al. (2012)* have recently shown that inhibitory IaINs are a major source of the tonic inhibitory drive underlying the push–pull control of motor activity in ankle extensor motor neurons. This tonic inhibition, in concert with tonic excitation, causes a net increase in the force modulation produced by ankle extensor muscles. Our observation that V1 and V2b INs are the sole source of Ia inhibition to motor neurons (*Zhang et al., 2014*) suggests that they make a major contribution to inhibitory push–pull conductances. As such, the biased innervation of motor neurons by V1 and V2b INs (*Figure 8*) could facilitate push–pull in limb motor neurons under a range of behavioral conditions. For example, in situations where V2b INs are being dynamically activated by cutaneous feedback, the V1 INs, or subsets thereof, might be tasked with providing tonic inhibition to motor neurons.

## Comparative analysis of V1 and V2b IN function in vivo vs in vitro

Our in vivo behavioral analyses showing an inhibitory network comprised of either V1 or V2b INs still generates an alternating pattern of flexor–extensor activity in awake behaving mice, albeit an abnormal one, concurs with our in vitro analysis showing flexor–extensor alternation is only completely degraded when both the V1 and V2b IN populations are inactivated (*Zhang et al., 2014*). These findings argue that the V1 and V2b INs can function in a redundant manner to produce a grossly alternating flexor–extensor locomotor output, and they are in general agreement with the mixed innervation of flexor and extensor motor neurons by V1 and V2b INs (*Figure 8*; *Zhang et al., 2014*).

The co-innervation of flexor and extensor motor neurons by V1 and V2b INs may contribute to the functional recovery that occurs following V1 or V2b IN ablation. This functional recovery is suggestive of a degree of plasticity in the inhibitory control of flexor–extensor movements by both inhibitory populations. Interestingly, the functional deficits that persist after ablating the V1 INs are more pronounced than those found following V2b IN ablation. This might be attributable to the relative abundance of these two cell types in the lumbar spinal cord, where there are twice as many V1 INs as compared to V2b INs (*Zhang et al., 2014*). Differences in the axonal morphology of V1 and V2b INs may also contribute to the persistent hindlimb hyperflexion phenotype V1 IN-ablated mice display, as the motor pools innervating hip and ankle flexors tend to be skewed rostrally in the lumbar spinal cord when compared to their extensor antagonist motor pools (*McHanwell and Biscoe, 1981*; *Yakovenko et al., 2002*). In this context, it is worth noting that V2b INs predominantly project their axons caudally (*Figure 8—figure supplement 1*) and may therefore have a limited capacity to substitute for those V1 INs that have rostrally projecting axons. In addition to this, there are fewer inhibitory contacts from V2b INs onto flexor motor neurons (*Figure 8*; *Zhang et al., 2014*). By contrast, the V2b IN-ablated mice regain a large measure of normal limb movement in the aftermath of V2b IN ablation. In this instance, V1 INs, many of which project caudally (*Figure 8—figure supplement 1*), may compensate for the initial loss of V2b IN-derived inhibitory inputs that are proportionately more abundant on extensor motor neurons.

The one locomotor deficit that persists when the V2b INs are removed is the delay in the transition from stance to swing during walking (*Figure 6*). This transition is controlled in part by Ib pathways from ankle extensor GTOs (*Duysens and Pearson, 1980*; *Pearson, 2008*). Our characterization of presynaptic inputs to V1 and V2b INs showing the V2b INs receive inputs from neurons in the dorsal horn (F Stam and MG, unpublished findings) is a strong indication that inhibitory IbINs are derived from the V2b IN population. As such, the presumed loss of inhibitory IbINs that occurs when V2b INs are ablated could account for the delayed transition from stance to swing that we observe (*Pearson, 2008*). The loss of IbINs in the V2b IN-ablated mice might also explain why these mice do not show any major change in GS and TA muscle activity during swimming, as Golgi tendon-derived sensory feedback is normally attenuated during this activity (*Akay et al., 2014*).

In summary, this study demonstrates that V1 and V2b INs differentially control flexor–extensor motor output, with V1 INs suppressing flexor motor activity during the stance/extension phase of walking to ensure proper extension of the limbs, while the V2b INs suppress extensor activity to facilitate limb extension and ensure the timely transition from stance to swing. Our results are consistent with the V1 and V2b INs contributing to the dynamic control of limb movements during walking via their differential effects on flexor and extensor motor activity. It should be noted that the V1 and V2b IN populations are made up of multiple cell types and the respective contribution that each of these cell types make to flexor–extensor-driven behaviors needs to be assessed. Future efforts to evaluate in depth how these two populations control flexor–extensor motor behaviors will require a better understanding of the molecular, anatomical, and physiological diversity that exists within these two inhibitory IN populations, coupled with a detailed functional characterization of V1 and V2b IN subtypes.

## Materials and methods

### Experimental procedures

#### Generation, genotyping and breeding of mice
All animal experiments were conducted according to NIH guidelines using protocols approved by the Salk Institute for Biological Studies IACUC. Animals were housed on a 12-hr light/dark schedule with ad lib access to food and water.

#### Generation of *Mapt*<sup>ds-DTR</sup> knockin mice
Sequences encoding for the simian diphtheria toxin receptor (DTR) were fused to EGFP and bGH polyA sequences, subcloned downstream of a modified double-stop (ds) cassette (*Kim et al., 2009*) and inserted into the *Mapt* locus (*Hippenmeyer et al., 2005*). The *Mapt*<sup>ds-DTR</sup> targeting vector was linearized and electroporated into 129S1/SvImJ 2A ES cells to derive the *Mapt*<sup>ds-DTR</sup> mouse knock-in line. Single stop variants of the *Mapt*<sup>ds-DTR</sup> knock-in allele were derived by crossing *Mapt*<sup>ds-DTR</sup> mice to *Actb::Flpe* (*Rodríguez et al., 2000*) or *Protamine::Cre* (*O'Gorman et al., 1997*) transgenic mice.

## Generation of Cdx2-FlpO transgenic mice

A 9.5-kb fragment containing the human *Cdx2* immediate promoter (*Hinoi et al., 2007*) followed by the β-globin intron, nlsFlpO, and a bGH-polyA expression cassette was cloned between tandem β-globin insulator element repeats. Transgenic mice were generated by standard pronuclei microinjection (Salk Institute Transgenic Core). Founder animals were identified by PCR and tested for FlpO-dependent recombination using *RC::Fela* and *RC::Fa* reporter mice (*Kim et al., 2009*). *Cdx2-FlpO* mice were maintained on a mixed C57BL/6-ICR background. Transgene expression remained stable over at least five generations.

## Genotyping

*Tau*ds-DTR knock-in animals were genotyped by PCR using TauEx2-forward (GTCAGATCACTAGACT-CAGCATCC) and TauEx2-reverse2 (GAATATTCAACCCCCTCGAA) primers to amplify a 376-bp fragment from the wt allele, and TauEx2-forward and pPFII-reverse (CGGCCTCGACTCTACGATAC) primers to amplify a 210-bp fragment from *Mapt*ds-DTR allele. *Cdx2-FlpO* transgenic mice were genotyped by PCR using FlpO-forward (TGAGCTTCGACATCGTGAAC) and FlpO-reverse (ACAGGGTCTTGGTCTTGGTG) primers. The genotyping of *En1*Cre and *Gata3*Cre mice has been described previously (*Sapir et al., 2004*; *Zhang et al., 2014*). Mice carrying the *Ai6, Ai14, RC::Fela,* and *R26*ds-HTB alleles were genotyped with Rosa4, Rosa10, and Rosa11 primers (*Zong et al., 2005*).

## Breeding

Quadruple-transgenic animals and controls were generated by crossing double heterozygous Cre/+; *Cdx2-FlpO*/+ males with *Mapt*ds-DTR/*Mapt*ds-DTR; *Ai14/Ai14* or *Mapt*ds-DTR/*Mapt*ds-DTR; *Ai6/Ai6* double homozygous females. In quadruple transgenic animals, DTR expression is restricted to the caudal postmitotic neurons, while the *Ai6 or Ai14* reporter is expressed in all Cre-descendent cells. Mice analyzed were from a mixed C57BL/6, 129/SV and ICR genetic background. Control animals were littermates lacking the *Cdx2-FlpO* allele.

## Diphtheria toxin administration

Diphtheria toxin (Sigma, St Louis, MO) was reconstituted in PBS. Adult mice (>8 weeks) were injected with a single dose, either i.p. (100 ng/g) or intrathecally (1 ng/g). For neonatal animals, P0–P5 pups were injected once s.c. (100 ng/g) in the nape of the neck. Lower doses (20 or 50 ng/g given at 10 day intervals) were used for ablating V2b INs in adults in order to increase their survival.

## Behavioral analysis

### Analysis of adult limb kinematics and electromyographic (EMG) activity

Hindlimb kinematic movements during walking or swimming and the associated muscle activity were analyzed by a high-speed video and simultaneous EMG recordings (*Pearson et al., 2005*). All recordings were performed at least 2 days after electrode implantation. Limb position and movements were tracked with custom-made three-dimensional light reflective markers (2 mm diameter) or black marks positioned on the iliac crest, hip, knee, ankle, paw, and tip of fourth digit of the left hindlimb. Fine EMG recording electrodes were manufactured and implanted into defined hindlimb muscles as previously described (*Pearson et al., 2005*). Mice were allowed to run or swim in a clear Plexiglas runway and their movements were recorded at 60 to 250 frames/s using an InLine high-speed digital camera (Fastec Imaging Corporation, San Diego, CA). EMG signals were amplified (DAM-50, WPI; DPA-2F, NPI) and recorded using the Clampex acquisition software (Molecular Devices). A time tag triggered by the camera allowed precise synchronization of the EMG and videos recordings.

The Clampfit data analysis module (Molecular Devices) was used for analyzing and graphing the EMG recording data. Video files were processed using the MaxTRAQ software package (Innovision Systems, Columbiaville, MI). The markers were identified and digitized manually or using the auto tracking feature. The position of the knee was calculated by triangulation based on known femur and tibia bone lengths using the plug-in MaxTRAQ VJR (virtual joint recognition). Data files were further analyzed in MaxMATE, a plug-in for Microsoft Excel. The positions, distances, angles, velocity were analyzed, and plots were generated for selected step sequence. Graphs of joint angle, joint velocity and the relative relationship of the angular changes during the step cycle were analyzed and stick figures representing the limb were generated to depict precisely the movement during walking using MaxMATE software.

## L-DOPA-induced airstepping

Early rhythmic motor activity was analyzed in 7- to 8-day-old neonatal mice using a drug-induced air stepping protocol (McEwen et al., 1997). Small black dots were made on the shoulder, elbow, wrist, knee, ankle, and paw to facilitate kinematic analysis. Locomotion was induced by injection of L-DOPA (100 µg/g body weight) subcutaneously into the nape of neck. Animals were immediately placed in a harness that supported the trunk while allowing free movement of the limbs. Airstepping typically commenced 10–15 min post-injection. High-speed video recordings (125 frames/s) of stable rhythmic locomotor activity were made over a 30-min period. Recordings of animals performing a stable sequence of stepping were selected for further kinematic analysis. Videos were digitized using MaxTRAQ software. To measure ankle muscle activity, two fine EMG recording electrodes were inserted into the TA and GS muscles of one hindlimb and EMG activity recorded during stable episodes of locomotion as described above.

## Optogenetic activation of V1 and V2b INs

Channelrhodopsin (ChR2) was excited with a blue LED light source (wavelength 470 nm, max. power 9–10 mW). Ventral root ENG recordings were performed as previously described (Lanuza et al., 2004). Following the induction of fictive locomotion rhythm by NMDA and 5-HT, 10 s pulses of light were delivered to the ventral surface of cord via an optic fiber cable (1 mm). Light excitation was delivered to the ventral surface of the lumbar cord at L1–L2, L3–L4, or L5–L6 levels to activate either the V1 or V2b INs.

## Histochemistry and imaging

Anesthetized animals were transcardially perfused with phosphate buffered saline (PBS) followed by 4% paraformaldehyde in PBS. Spinal cords were harvested, immersion fixed at 4˚C overnight and cryoprotected in 20% sucrose-PBS. β-galactosidase stainings and immunostainings were performed as previously described (Lanuza et al., 2004; Sapir et al., 2004). The following primary antibodies were used: rabbit anti-GFP (Molecular Probes), rabbit anti-Ds-Red (Clontech), anti β-galactosidase (Abcam), rabbit anti-Calbindin (Swant), goat anti-ChAT (Millipore), guinea pig anti-Chx10 (gift from Dr. Sam Pfaff), anti-CD86 (Biolegend), guinea pig anti-vGAT (Synaptic Systems), guinea pig anti-GlyT2 (Chemicon). Cell nuclei were stained and visualized with DAPI. Images were captured using a Zeiss LSM510 confocal microscope.

## Statistical analysis

All results were compared using the Student's $t$-test. All data are expressed as mean $\pm$ s.d. The threshold for significant difference was set at 0.05.

# Additional information

## Funding

| Funder | Grant reference | Author |
| --- | --- | --- |
| National Institutes of Health (NIH) | NS037075, NS090919 | Martyn Goulding |
| Canadian Institutes of Health Research (Instituts de recherche en santé du Canada) | CIHR MOP 86470 | Simon Gosgnach |

The funders had no role in study design, data collection and interpretation, or the decision to submit the work for publication.

## Author contributions

OB, Conception and design, Acquisition of data, Analysis and interpretation of data, Drafting or revising the article; JZ, KSG, SG, Acquisition of data, Analysis and interpretation of data; JD, Acquisition of data; JCK, SD, Contributed unpublished essential data or reagents; MG, Conception and design, Analysis and interpretation of data, Drafting or revising the article

## Ethics

Animal experimentation: This study was performed in strict accordance with the recommendations in the Guide for the Care and Use of Laboratory Animals of the National Institutes of Health. All of the animals were handled according to the approved institutional animal care and use committee (IACUC) protocol (06-013) of The Salk Institute for Biological Studies. All surgeries were performed

under isofluorane anesthesia. Every attempt was made to reduce pain and suffering, including NSAID administration when necessary.

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
