## [Decision Letter]

Thank you for sending your work entitled “A genetically-defined asymmetry underlies the inhibitory control of flexor-extensor locomotor movements” for consideration at *eLife*. Your article has been favorably evaluated by Eve Marder (Senior Editor) and three reviewers, one of whom, Ronald L. Calabrese, is a member of our Board of Reviewing Editors.

The Reviewing Editor and the other reviewers discussed their comments before we reached this decision, and the Reviewing Editor has assembled the following comments to help you prepare a revised submission.

The authors present a very thorough analysis of the physiological roles of V1 and V2b classes of interneurons in the spinal cord of mice. They use elegant genetic techniques to ablate separately these two classes of interneurons from the subcervical spinal cord only in response to DTX injection. They assess the effects of these ablations with a battery of behavioral and electromyographic assays all in awake behaving mice (mainly neonatal). They find that V2b INs predominantly inhibit extensor motor neurons and cause hyperextension when ablated, whereas V1 INs predominantly inhibit flexor motor neurons and cause hyperflexion when ablated. They supplement this work and support these findings with optogenetic experiments in isolated neonatal spinal cord. They then perform anatomical analysis of contacts between each of these two classes of INs and motor neurons and show convincingly that V2b INs contact mainly but not exclusively extensor motor neurons and V1 INs contact mainly but not exclusively flexor motor neurons. Thus inhibitory innervation of motor neurons by these two classes of INs is differential but push-pull. These findings have very important implications for the functional organization of spinal motor circuits. The data set is extensive and thorough. The figures are clear and show necessary data; supplemental figures are extensive but useful. This paper will provoke wide interest among the readers of *eLife*.

There were some concerns that the authors should address in revision. Each of the reviewers come from different perspectives but provided complementary guidelines for revisions:

1) The context of the present work needs to be broadened and deepened. Reviewer 2 has several specific suggestions about spinal cord research that should be considered mainly in the Introduction and Discussion. These are repeated in their entirety and should be addressed; it may be that not all the suggested citations are necessary but many can be incorporated to the benefit of the manuscript and each general point should be considered in the authors' response:

In the Introduction, the references to Feldman and Orlovsky, and Pratt and Jordan are fine, but given Berg et al.'s hypotheses on a simultaneous shift in excitation and inhibition – including the IaIN (the Hultborn's group in Copenhagen used the “Jankowska method” of simultaneous recording of IaIN and the unitary IPSPs in the target motoneurons, thus 100% identified, during fictive locomotion – those interneurons indeed alternate with excitation and are partly responsible for the hyperpolarized phase (please see Hultborn et al., Reciprocal Ia inhibition contributes to motoneuronal hyperpolarisation during the inactive phase of locomotion and scratching in the cat., J Physiol., 2011 Jan 1;589(Pt 1):119-34).

It is true that Ib inhibitory interneurons are well characterized, but they are largely inhibited during locomotion. Instead, a disynaptic and polysynaptic excitation is seen. There are five major references:

[13]. Proprioceptive input resets central locomotor rhythm in the spinal cord., Experimental Brain Research, 68, 643-656.

Pearson et al., (1993). Reversal of the influence of group Ib afferents from plantaris on activity in medial gastrocnemius muscle during locomotor activity., Journal of Neurophysiology, 70, 1009-1017.

[24], Transmission in a locomotor-related group lb pathway from hindlimb extensor muscles in the cat., Experimental Brain Research, 98, 213-228.

Angel et al. (1996), Group I extensor afferents evoke disynaptic EPSPs in cat hindlimb extensor motorneurones during fictive locomotion., J Physiol., Aug 1;494 ( Pt 3):851-61.

[4], Candidate interneurones mediating group I disynaptic EPSPs in extensor motoneurones during fictive locomotion in the cat., J Physiol., Mar 1;563(Pt 2):597-610.

In the Introduction you state: “The latter study revealed […] when both of these classes of neurons are inactivated.” As it is formulated here, it is virtually impossible not to cite the following: Talpalar et al. (2011), Identification of minimal neuronal networks involved in flexor-extensor alternation in the mammalian spinal cord., Neuron 71, 1071-1084.

The above is based on the model presented by Miller et al. (1977), The spinal locomotor generator., Exp. Brain Res. 30, 387-403.

Also, it would be helpful to the reader to mention which interneuronal groups belongs to V1 and V2b groups, as far as it is known. In the Results section and especially in the Discussion, it is almost “forgotten” that the Renshaw cells belong to the V1 group – most of the discussion focuses on Ia INs being. They are belonging both to the V1 and V2b groups – perhaps biased in relation to where they are projecting.

In the Results section, are the authors thinking of “rhythm generator” or “pattern generator” – the multilayered model? Speaking about *rhythm* makes the reader wonder if that is part of the authors thinking or not – the issue is not really raised anywhere.

In the Talpar et al. paper it is really demonstrated beyond doubt that Ia inhibitory interneurons and Renshaw cells CAN produce a rhythm under special circumstances – to which extent they do contribute in a normal animal is unknown – but blockade of Renshaw cells does not affect the rhythm ([54], The role of Renshaw cells in locomotion: antagonism of their excitation from motor axon collaterals with intravenous mecamylamine., Exp Brain Res., 66(1):99-105.)

Regarding the strong inhibition of motoneurons by Renshaw cells, the reference to Windhorst may be OK, but the following recent publication is a must: [8], The recurrent case for the Renshaw cell., J Neurosci., Sep 17;34(38):12919-32.

Reviewer 3 is concerned that the paper overlooks the literature in non-mammalian preparations where many of the suggested mechanisms have been studied (*Xenopus*, lamprey, zebrafish). There are recent data in *Xenopus* and zebrafish about the role of V1 interneuron function as well as the functional diversity of specific classes of interneurons (V0 and V2a) and their connectivity in zebrafish.

2) The paper creates ambiguity about the role of these two classes of INs in motor patterning versus motor rhythm generation. The IN classes seem to affect flexor extensor balance (patterning) but not to disrupt the production of rhythmic motor bursts (rhythm generation) which is controlled by spinal oscillatory timing networks. In this conception CPG embraces both patterning and rhythm generating networks and thus CPG is an ambiguous term when trying to specify the exact role of a particular class of IN. The IN classes appear to affect cycle period somewhat in intact mice but this could be mediated by sensory feedback; the optogenetic experiments on fictive locomotion in Figure 7 could partially disambiguate this effect on period but a definitive answer would require monitoring rhythm generating elements centrally. There is no apparent strong effect on period in the figure, so perhaps the V1 and V2b INs do not have access to the rhythm generator.

3) In Figures 4 and 5, the authors need to explain the apparent discrepancy between the lack of movements around the ankle joint (Figure 4) and the occurrence of alternation EMG activity between GS and TA (Figure 5) after V1 and V2b interneuron ablation. The changes in EMG activity are used to explain the effects seen on the movements of the ankle joint, but this activity does not fully account for the executed movements. Is something missing here?

4) Figure 6, the effect of V1 ablation is different between walking and swimming. While the EMG activity is alternating during walking, it becomes synchronous during swimming. This needs some explanations to put it in some functional context. The quantification of the phase relationship between GS and TA in all conditions should be provided.

---

## [Author Response]

*1) The context of the present work needs to be broadened and deepened. Reviewer 2 has several specific suggestions about spinal cord research that should be considered mainly in the Introduction and Discussion. These are repeated in their entirety and should be addressed; it may be that not all the suggested citations are necessary but many can be incorporated to the benefit of the manuscript and each general point should be considered in the authors' response. […] Reviewer 3 is concerned that the paper overlooks the literature in non-mammalian preparations where many of the suggested mechanisms have been studied (*Xenopus*, lamprey, zebrafish). There are recent data in* Xenopus *and zebrafish about the role of V1 interneuron function as well as the functional diversity of specific classes of interneurons (V0 and V2a) and their connectivity in zebrafish*.

The Introduction of the manuscript has been substantially rewritten. There is now a much more in depth discussion of inhibitory neuron cell types in the cat and what is known about their activity during locomotion and their potential contribution to flexor-extensor motor behaviors. The papers suggested by Reviewer 2 that describe the role of Ib sensory feedback are cited and discussed, together with an additional study by Shoji et al. that describes Ib inhibition in humans during walking. We have included the additional references on Renshaw cells suggested by reviewer 2 (54, 8), and a further reference to the study by [53]. We decided not to cite Miller and Scott, 1977 as well as Talpalar et al. 2011. The Miller and Scott model has been widely discounted by a number of studies including that of Talpalar et al., and we feel that any discussion of it would not be informative, and instead might be confusing to those outside of the field. We also describe what is known about the cell types that make up the V1 and V2b IN populations. Overall, we feel that this has resulted in a much more thoughtful and scholarly introduction.

*2) The paper creates ambiguity about the role of these two classes of INs in motor patterning versus motor rhythm generation. The IN classes seem to affect flexor extensor balance (patterning) but not to disrupt the production of rhythmic motor bursts (rhythm generation) which is controlled by spinal oscillatory timing networks. In this conception CPG embraces both patterning and rhythm generating networks and thus CPG is an ambiguous term when trying to specify the exact role of a particular class of IN. The IN classes appear to affect cycle period somewhat in intact mice but this could be mediated by sensory feedback; the optogenetic experiments on fictive locomotion in*
Figure 7
*could partially disambiguate this effect on period but a definitive answer would require monitoring rhythm generating elements centrally. There is no apparent strong effect on period in the figure, so perhaps the V1 and V2b INs do not have access to the rhythm generator*.

In our discussion of the deletions that occur during airstepping (Figure 5), we now make it clear that the deletions that we see are likely to be non-resetting in nature, which is consistent with the V1 and V2b Ins operating at the level of pattern formation rather than rhythm generation. We have a modified the Discussion section to tone down our comments about these deletions, as this was not the major focus of our study.

*3)*
Figures 4 and 5*: The authors need to explain the apparent discrepancy between the lack of movements around the ankle joint (*Figure 4*) and the occurrence of alternation EMG activity between GS and TA (*Figure 5*) after V1 and V2b interneuron ablation. The changes in EMG activity are used to explain the effects seen on the movements of the ankle joint, but this activity does not fully account for the executed movements. Is something missing here*?

We feel that the EMG activity that we see in Figure 5 does largely account for the limited movement that occurs around the ankle joint. For example in comparing control and V1 IN-ablated mice, what one sees is a marked prolongation of TA activity. Admittedly there are short rhythmic bursts of GS muscle activity, however, these are severely truncated and in some instances are partially co-active with TA muscle activity. We would like to suggest that these short bursts of ankle extensor activity are insufficient for the full opening of the ankle joint. There is some opening, but this is rather limited. Likewise in the V2b IN-ablated animals we see very long bursts of GS extensor activity and a marked reduction in the duration and amplitude of TA activity, which is again consistent with there being very little movement around the ankle joint.

*4)*
Figure 6*: The effect of V1 ablation is different between walking and swimming. While the EMG activity is alternating during walking, it becomes synchronous during swimming. This needs some explanations to put it in some functional context. The quantification of the phase relationship between GS and TA in all conditions should be provided*.

We discuss in more detail the differences in ankle extensor and flexor EMG activity that occur during walking versus swimming in the V1 IN-ablated mice (see the third paragraph of the subsection “Deficits in flexor-extensor motor coordination persist following V1 and V2b IN ablation”). We believe that these differences are likely to be due to reduced Ib sensory feedback, which is known to promote extension and also strongly influence the transition from stance to swing during walking. In the absence of V1 inhibitory IN-mediated sensory feedback, attenuation of this pathway during swimming might remove the Ib sensory signals that suppress flexor activity during extension, as along with timing signals that are provided by Golgi tendon activity and possibly also cutaneous signals from the plantar surface of the foot. This point is raised again in the Discussion. The phase relationships between GS and TA are now provided in Figure 6—figure supplement 1.